# Anti-Influenza with Green Tea Catechins: A Systematic Review and Meta-Analysis

**DOI:** 10.3390/molecules26134014

**Published:** 2021-06-30

**Authors:** Anchalee Rawangkan, Kirati Kengkla, Sukrit Kanchanasurakit, Acharaporn Duangjai, Surasak Saokaew

**Affiliations:** 1School of Medical Sciences, University of Phayao, Phayao 56000, Thailand; ewmedsci@gmail.com (A.R.); achara.phso@gmail.com (A.D.); 2Unit of Excellence on Clinical Outcomes Research and IntegratioN (UNICORN), School of Pharmaceutical Sciences, University of Phayao, Phayao 56000, Thailand; 3Division of Pharmacy Practice, Department of Pharmaceutical Care, School of Pharmaceutical Sciences, University of Phayao, Phayao 56000, Thailand; kirati.ke@gmail.com (K.K.); sukrit.ka@up.ac.th (S.K.); 4Center of Health Outcomes Research and Therapeutic Safety (Cohorts), School of Pharmaceutical Sciences, University of Phayao, Phayao 56000, Thailand; 5Unit of Excellence on Herbal Medicine, School of Pharmaceutical Sciences, University of Phayao, Phayao 56000, Thailand; 6Division of Pharmaceutical Care, Department of Pharmacy, Phrae Hospital, Phrae 54000, Thailand; 7Biofunctional Molecule Exploratory Research Group, Biomedicine Research Advancement Centre, School of Pharmacy, Monash University Malaysia, Bandar Sunway, Selangor Darul Ehsan 47500, Malaysia; 8Novel Bacteria and Drug Discovery Research Group, Microbiome and Bioresource Research Strength, Jeffrey Cheah School of Medicine and Health Sciences, Monash University Malaysia, Bandar Sunway, Selangor Darul Ehsan 47500, Malaysia

**Keywords:** EGCG, green tea catechins, influenza, meta-analysis

## Abstract

Influenza is one of the most serious respiratory viral infections worldwide. Although several studies have reported that green tea catechins (GTCs) might prevent influenza virus infection, this remains controversial. We performed a systematic review and meta-analysis of eight studies with 5048 participants that examined the effect of GTC administration on influenza prevention. In a random-effects meta-analysis of five RCTs, 884 participants treated with GTCs showed statistically significant effects on the prevention of influenza infection compared to the control group (risk ratio (RR) 0.67, 95% CIs 0.51–0.89, *p* = 0.005) without evidence of heterogeneity (*I*^2^ = 0%, *p* = 0.629). Similarly, in three cohort studies with 2223 participants treated with GTCs, there were also statistically significant effects (RR 0.52, 95% CIs 0.35–0.77, *p* = 0.001) with very low evidence of heterogeneity (*I*^2^ = 3%, *p* = 0.358). Additionally, the overall effect in the subgroup analysis of gargling and orally ingested items (taking capsules and drinking) showed a pooled RR of 0.62 (95% CIs 0.49–0.77, *p* = 0.003) without heterogeneity (*I*^2^ = 0%, *p* = 0.554). There were no obvious publication biases (Egger’s test (*p* = 0.138) and Begg’s test (*p* = 0.103)). Our analysis suggests that green tea consumption is effective in the prophylaxis of influenza infections. To confirm the findings before implementation, longitudinal clinical trials with specific doses of green tea consumption are warranted.

## 1. Introduction

Influenza is one of the world’s greatest infectious disease challenges. Annual epidemics result in an estimated 3 to 5 million cases of severe illness and approximately 290,000 to 650,000 deaths from respiratory illnesses, particularly during the global winter [1]. Influenza viruses of subtypes A(H1N1) and A(H3N2) are currently circulating in humans. The influenza type B viruses currently circulating belong to either the B/Yamagata or the B/Victoria lineages. The virus particles can be transmitted from person to person by respiratory droplets, aerosols, and contact [2,3]. Various hygiene and physical distancing measures have helped to reduce the transmission of the influenza virus [4], and while the most effective method of preventing the disease is vaccination, it is not 100% effective. Therefore, sustained preventive measures are needed to prevent influenza infection.

Various herbal remedies have been used to prevent and treat viral respiratory diseases. Those found to be effective include ginseng and North American ginseng, elderberry, maoto, licorice roots, antiwei, berries, echinacea, plants containing carnosic acid, pomegranate, guava tea, and bai shao [5,6,7,8]. Accumulating evidence confirms the therapeutic effect of polyphenols in various models of influenza virus infection, suggesting that polyphenol-rich plants may be considered a new natural source for the development of future anti-influenza drugs [9]. One of these polyphenol-rich plants is green tea.

Green tea is a daily drink consumed throughout the world. One cup (120 mL) of brewed green tea provides an estimated 150 mg of total catechins, which include the four main catechins: 10–15% (−) epigallocatechin gallate (EGCG), 6–10% (−) epigallocatechin (EGC), 2–3% (−) epicatechin gallate (ECG), and 2% (−) epicatechin (EC) [10]. 

Green tea is a source of amino acids, proteins, and lipids, as well as rich in essential chemical compounds such as trace elements and vitamins A, B, C, E, and K [11]. Green tea catechins (GTCs) and EGCG are now receiving considerable attention as having antitumor, antioxidant, anti-inflammatory, anti-diabetes, anti-obesity, anti-hypertension, and anti-microbial infection effects [11,12]. Various evidence suggests that GTCs are an effective antiviral agent, e.g., against influenza and SARS-CoV-2 viruses [13,14,15]. The stability and bioavailability of GTCs and EGCG as viral preventatives are still unclear. Normally, EGCG reaches a maximum of 1 μM in plasma within 2 h after drinking green tea and is excreted from the blood with a half-life of 5 h in humans. EGCG is mainly present in free form at about 80% in plasma, while other catechins are strongly conjugated with glucuronic acid and/or sulfate groups [16]. In rodent studies, EGCG has been found to be distributed in a variety of target organs, such as the digestive tract, blood, brain, liver, kidney, and spleen. EGCG is excreted into feces via bile, while EGC, ECG, and EC are excreted via both bile and urine [17,18]. GTCs are metabolized by hepatic and intestinal enzymes and the normal flora of intestinal microorganisms for glucuronidation, sulfation, methylation, and ring cleavage. The bioavailability of orally administered EGCG-enriched GTC capsules after overnight fasting results in the highest EGCG exposure in plasma [17,19]. However, EGCG bioavailability shows high interindividual variability related to gastrointestinal absorption, stability of the molecule, nutritional environment, and administration conditions. It is important to note that green tea catechins are more stable at low temperatures and acidic pH conditions [20,21,22]. An in vitro study showed that EGCG catechins prevented influenza virus replication by blocking hemagglutinin (HA) in the adsorption phase [23,24,25]. The hemagglutination inhibition assay (HI) showed that the galloyl group at the 3-position of EGCG and ECG catechins binds to the HA spike on the influenza virus envelope, leading to the inhibition of the attachment activity of the viral HA and sialic acid receptor on red blood cells, and then acting on the acidification of the intracellular compartments of endosomes and lysosomes in the penetration and uncoating phases [26]. EGCG also suppressed viral RNA synthesis by inhibiting viral endonuclease activity [27], and finally blocked the viral progeny release phase by inhibiting influenza neuraminidase activity, which is similar to the mechanism of action of oseltamivir, an active anti-influenza drug [28].

Several clinical studies have been conducted on the anti-influenza effects of catechins following the consumption of green tea via different routes of administration, i.e., drinking brewed green tea, taking green tea capsules as a dietary supplement, or gargling with green tea. Of these, the clinical trials had some limitations, i.e., the size of the population, duration of the study, and the possibility of bias due to residual confounding and unadjusted confounders (occupation, lifestyle including diet, immune function, chronic diseases, etc.) [29]. Therefore, the clinical effects of catechins for the prevention of influenza have remained unclear.

A previous meta-analysis review suggested that green tea catechins may have preventive effects on influenza, but these studies had limitations since they only used green tea as a gargle rinse [30,31]. Although several clinical evidence studies have reported that GTCs might prevent influenza virus infection, this remains controversial. Therefore, we performed a systematic review and meta-analysis of randomized controlled trials and prospective cohort studies to evaluate the effect of green tea catechin consumption on the prevention of influenza infection.

## 2. Results

### 2.1. Literature Search

A PRISMA flow diagram is shown in Figure 1. Qualitative and quantitative synthesis searches identified a total of 681 articles (250 articles from Scopus, 103 articles from PubMed, 325 articles from EMBASE, but none found in the Cochrane Central Register of Clinical Trials), and an additional three articles were identified by a review of reference lists. We found 516 articles after removing duplicates and unrelated studies by screening titles and abstracts manually and using Rayyan, an online systemic application. In this way, 12 articles were identified as eligible studies during full-text reading. Unfortunately, four articles were excluded due to the unknown number of participants (*n* = 2) and inappropriate outcome assessment, i.e., evidence of bacterial infection (*n* = 2). Finally, eight studies were included in the analysis. 

### 2.2. Characteristics of the Included Studies

The main characteristics and results of the included articles are summarized in Table 1. The eight selected studies included five RCTs, with a study duration of 3–5 months [32,33,34,35,36], and three prospective cohort studies, with a study duration of 3–8 months [37,38,39]. The 5048 participants included were divided into the group taking GTCs (3107 participants) and the group with no treatment (1941 participants). Most of the participants were under 65 years old (there was an age range of 6 to 83 years). Half of them had received the seasonal influenza vaccine. The route of administration of GTCs was by gargling in four studies [33,34,35,39] by taking GTCs in capsule form in two studies [32,36], and by drinking in two studies [38,39]. The concentration of GTCs taken per day varied from approximately 100 mg to 280 mg for gargling. On the other hand, subjects were randomly assigned to take GTC capsules containing 378 and 1500 mg/day or to consume green tea by drinking 1–5 cups/day or at least 2 cups/week containing 137–685 mg of GTCs (Table 1).

### 2.3. Quality Assessment

Each RCT was confidential and considered to be low risk, as the evaluation of six domains in the Cochrane Risk of Bias (RoB 2.0) tool confirmed. In terms of quality assessment by the RoB 2.0, two studies had a low risk of bias [33,34], one study had a moderate risk of bias [32], and the other two exhibited a serious risk of bias [35,36]. For quality evaluation through NOS, studies were considered high quality if they received a score of 7 stars or more. In this analysis, two studies received 835 or 9 stars [37], with the remaining low quality study receiving 6 stars [38]. However, according to ROBINS-I, one study was of the overall critical quality (Appendix A). Details of the quality assessment by the RoB 2.0 and NOS tools are presented in Figure 2 and Table 2, respectively.

### 2.4. Clinical Effects of GTCs on Influenza Prevention

The incidence of influenza infection was defined on the premise of fever (temperature ≥37.8 °C) and the following clinical signs: chills, cough, sore throat, headache, myalgia, malaise, rhinorrhea, and loss of appetite or diarrhea. Influenza infection was confirmed by laboratory diagnostics, i.e., the detection of viral antigens by an immunochromatographic assay, a rapid antigen detection test commonly used in clinical practice to define influenza virus type A or B [32,34,35,36,38,39], or cases were confirmed by reverse transcription polymerase chain reaction (RT-PCR), as for the influenza A subtype H1N1 pandemic 2009 (H1N1pdm09). Multiplex PCR was also performed to identify co-infections of seasonal influenza, an influenza A (H3N2) and influenza B, with H1N1pdm09 (Appendix A) [37].

Prior to data analysis, heterogeneity among the five RCTs and three cohort studies was assessed (Figure 3 and Table 3). The overall *I*^2^ statistic of both study designs suggested an absence of heterogeneity (*I*^2^ = 0%; *p* = 0.554); RCTs, *I*^2^ = 0%, *p* = 0.629, and for the cohort studies, *I*^2^ = 3%, *p* = 0.358. A forest plot (Figure 3) of the 3,107 participants treated with GTCs showed statistically significant effects on the prevention of influenza infection compared to those in the control group, as evidence by a pooled RR of 0.67 (95% CIs 0.51–0.89) for the five RCTs, and an RR of 0.52 (95% CIs 0.35–0.77) for the three cohort studies. We then examined the overall effect in a subgroup analysis of the mode of administration, gargle rinse or orally ingested items (taking capsules and drinking). The results of a pooled RR model for the eight studies are shown in Figure 4.

Taking capsules: two studies [32,34] estimated the effect of taking GTC capsules on preventing influenza infection. The overall effect of the pooled RRs demonstrated a significantly reduced risk of influenza (RRs, 0.54; 95% CIs, 0.26–1.13, *p* = 0.003) with a median level of heterogeneity (*I*^2^ = 49%, *p* = 0.161). 

Gargling: four studies [33,35,36,37] assessed the effects of daily gargling with GTCs. The summary effect of the pooled RRs was 0.70 (95% CIs 0.44–1.09, *p* = 0.069) with very low evidence of statistical heterogeneity (*I*^2^ = 2%, *p* = 0.381). 

Drinking: the studies presented in [38,39] showed that green tea consumption was significantly associated with influenza prevention. The effect of the pooled RRs was 0.54 (95% CIs 0.37–0.80, *p* = 0.002) without evidence of heterogeneity (*I*^2^ = 0%, *p* = 0.636). 

### 2.5. Sensitivity and Subgroup Analyses

The results of the sensitivity and subgroup analyses are shown in Table 3. For sensitivity and subgroup analysis, the data were stratified by study design. The RRs of RCTs were 0.67 (95% CIs, 0.51–0.89) for a random-effects model and 0.66 (95% CIs, 0.50–0.88) for a fixed-effects model, while the RRs of cohort studies were 0.49 (95% CIs, 0.33–0.73) for a random-effects model and 0.52 (95% CIs, 0.35–0.77) for a fixed-effects model. The sensitivity analysis was then performed after eliminating studies with a high risk of bias [35,36] or low quality of evidence [38]. After removal, the results showed that green tea was still significantly protective against infection caused by the influenza virus. It is important to note that the adverse events occurred in high dose consumption subjects [32]. From toxicological and human safety data, a safe intake level of 338 mg EGCG/day for adults was derived from tea preparations taken as a solid dosage form, and 704 mg EGCG/day could be considered for tea preparations in the beverage form [40]. In the subgroup analysis, results were stratified by EGCG dose, with a cut-off of 338 mg/day.

We found that the effect estimates from such an analysis still showed a trend toward the prevention of influenza infection: RR 0.66, 95% CIs 0.44–1.01, and RR 0.52, 95% CIs, 0.35–0.77, for RCTs and cohort studies, respectively (Table 3). At the same time, the route of administration of GTCs was also analyzed. The results showed that all routes of administration (gargling, taking GTC capsules, and drinking) showed efficacy in preventing influenza. The incidence of clinically defined influenza virus infection showed that type A was more common than type B (Appendix A), although a proportion of participants had received a seasonal influenza vaccine. It is important to note that influenza vaccination in the group consuming green tea seemed to better prevent influenza infection. For the RCTs, the RR of participants with and without influenza vaccination was 0.36 (95% CIs 0.15–0.90) and 0.72 (95% CIs 0.54–0.96), respectively. In the cohort studied, the RR of participants with and without influenza vaccination was 0.13 (95% CIs 0.15–0.90) and 0.54 (95% CIs 0.54–0.96), respectively. The participants in these studies were mixed populations with divided age groups. Participants who were over 65 years old were assigned to the elderly group. This suggests that green tea consumption could be effective for any age group in the studied cohort (RR 0.13, 95% CIs 0.02–1.05 and RR 0.54, 95% CIs 0.37–0.8, respectively), regardless of gender (Table 3).

### 2.6. Publication Bias of Included Studies

The potential publication bias of the eight included studies is illustrated by a funnel plot (Appendix A). As only a small number of articles were included in this study, the visual translation of the plot may be uncertain. However, Begg’s test and Egger’s test analyses showed no significant difference (*p* = 0.138 and *p* = 0.103, respectively).

## 3. Discussion

We conducted a systematic review and meta-analysis to evaluate the possibility that green tea catechins prevent influenza infection. We found that participants with a history of green tea consumption showed a significant association with improved influenza prevention, with a small degree of heterogeneity. Based on these findings, it appears that consuming 137–685 mg of green tea catechin per day (1–5 cups), taking 378–1500 mg of supplement capsules per day, or gargling 100–280 mg per day is an ideal catechin level for influenza prevention. However, the EGCG concentration of green tea preparations should be stated as the output to avoid hepatotoxicity. Based on 1 g leaf/100 mL infusion, brewed green tea contains an average of 126.6 mg total catechins and 77.8 mg EGCG per 100 mL as consumed, according to the United States Department of Agriculture (USDA) Flavonoid Database. As a result, each 240 mL serving of brewed green tea could contain 304 mg total catechins, including 187 mg EGCG. As a result, for people who drink three 8 oz. cups of green tea per day, daily catechin and EGCG intakes are reported to be around 912 and 560 mg, respectively [41]. A systematic review of adverse event data from 159 human studies revealed results consistent with toxicological findings that a limited range of concentrated, catechin-rich green tea preparations resulted in hepatic adverse events in a dose-dependent manner when taken in large bolus doses, but not when consumed as brewed tea or extracts in beverages or as part of the diet [40]. In rare cases, adverse events such as flatulence, stomach upset, dizziness, rash, and constipation have been reported when taking supplement capsules with EGCG concentrations greater than 338 mg/day [32]. This should be considered.

### 3.1. Comparison with Other Studies

The analysis of each route of administration of GTCs (gargling, taking GTC capsules, or drinking) showed efficacy in preventing influenza. Previously, a meta-analysis of gargling with green and black tea to prevent influenza infection was reported by Ide’s group [31]; the data from the RCTs were then reanalyzed by the same research group using Bayesian approaches [30]. The results suggested that gargling with green tea may slightly reduce influenza compared to gargling with water, which is similar to this study (RR, 0.75 (95% CIs 0.48–1.18) of RCTs events). Consequently, regular gargling with GTCs is a choice for preventing influenza-like infections for those who dislike drinking brewed green tea. On the other hand, drinking green tea or taking GTC supplements that contain catechin active compounds at two and five times the concentration of gargling solution are also able to significantly prevent influenza infection. However, the number of studies is very small: two cohorts of drinking and two RCTs of taking capsules. Therefore, further studies are needed to confirm this finding. In addition, RCTs have shown that drinking green tea catechins prevents infection not only by the influenza virus, but also by other upper respiratory infectious viruses such as respiratory syncytial virus, adenovirus, and rhinovirus [42,43].

The mechanism of action of GTCs against influenza viruses in an in vitro study has already been mentioned. In an in vivo and ex vivo study, EGCG and other catechins were shown to activate T cell function by increasing the IL-12 and IFN-γ secretion to defend and respond to influenza infection and also enhance the antigenic challenge with an anti-inflammatory effect [32,44]. Moreover, not only catechin but also theanine, a non-protein amino acid of green tea, might be involved in the accumulation of T cell functions [34]. Nevertheless, the bioavailability of catechins via oral administration may be individually important for a therapeutic effect [45]. Gargling involves rinsing the mouth and throat with a liquid that is kept moving by breathing through it with a gurgling sound [46]. Although there are reports suggesting the prevention of influenza by gargling, it should be noted that the mechanism of prevention of these respiratory infections by gargling is unclear. Therefore, a more detailed study is needed to verify the effects of the prevention of influenza and whether this is a direct effect of gargling itself or the effect of catechin.

### 3.2. Clinical Implication

The highest incidence of clinically defined influenza virus infection is reported for type A. This is consistent with the WHO finding that only type A influenza viruses are known to cause pandemics [1]. In a prospective household cohort study by Delabre’s group, pandemic influenza A/H1N1pdm09 was found to be co-infected with seasonal influenza A (H3N2) and influenza B in 68% (30/44) and 27% (10/37), respectively [39]. Participants who were vaccinated with the seasonal influenza vaccine before participating in the study could also contribute to influenza prevention. It is important to note that the influenza vaccine seems to be more effective in preventing influenza infection in the group that consumed green tea. Seasonal influenza vaccination is especially important for people who are at high risk of influenza complications and for people who live with or care for people at high risk, such as pregnant women, children aged 6 months to 5 years old, elderly people (over 65 years of age), people with chronic illnesses, and healthcare workers [47]. Therefore, future studies need to clarify the potential for catechins to serve as an effective influenza vaccination and should be performed for the evaluation of its effectiveness in longitudinal clinical trials.

### 3.3. Strengths and Limitations 

The strengths of this study should be highlighted as follows. We conducted a comprehensive search of four major databases (PubMed, SCOPUS, Embase, and Cochrane Library), which is a standard method for conducting a systematic review without restrictions on language or study design. We performed a rigorous assessment of the methodological quality of the included studies, as determined by the ROBINS-I and NOS. The results showed consistency in terms of the quality of the studies included. This meta-analysis adhered to the standard methodology of systematic reviews and meta-analyses as required by the PRISMA checklist (see Appendix A). Finally, our study covered the current evidence and was conducted using appropriate statistical methods for the analyses.

However, our study also had some limitations. Because present studies do not show publication bias suggesting that green tea consumption may prevent influenza, a smaller number of clinical trial reports may have been affected by the analysis. Therefore, we suggest that consensus needs to be established or confirmed on a larger scale with more detailed instructions in future studies, especially in RCTs investigating daily green tea consumption. It is important to note that the comparison of each mode of green tea consumption, i.e., drinking, gargling, or taking a dietary supplement with GTCs, was not completed in this study due to the limited number of subjects. Moreover, the effect remains inconclusive in the more sensitive groups, i.e., the ingredients and formulation of the intervention. The cultivation area and extraction process of green tea are related to the quality and stability of active catechin content, including brewing conditions, sterilization, pH, storage time, and temperature [48,49,50]. The home environment, social contacts, and use of public transportation have a high risk of exposing individuals to influenza viruses. Asthma and chronic obstructive pulmonary disease have been identified as possible risk factors for influenza infection [47,51]. In addition to preventing influenza by consuming green tea, personal hygiene is important for attenuating influenza. Hand hygiene and face masks appear to prevent influenza virus transmission when performed within 36 h of the onset of symptoms in the patient [52,53,54].

Finally, the results of this report can provide an additional measure for influenza prevention, and could be considered an alternative option to a pharmaceutical strategy. Regarding the oral administration of EGCG, this reduced the mortality rate of influenza infection, which was equivalent to the oral administration of oseltamivir in mice [55]. Consequently, EGCG treatment alone or in combination with therapy with anti-influenza drugs may be used as a treatment of choice for influenza drug-resistant variants, which have been frequently reported [56].

## 4. Materials and Methods

This study was performed according to the Preferred Reporting Items for Systematic Reviews and Meta-Analyses (PRISMA) statement [57]. This study was registered on PROSPERO (registration number: CRD42020176371) [58]. 

### 4.1. Search Strategies

We searched PubMed, Scopus, Embase, and the Cochrane Central Register of Clinical Trials for relevant original research articles published from inception up to February 2021. The search strategy was carried out with the following keywords: “green tea,” “catechin,” “epigallocatechin gallate,” “influenza,” “flu,” and “common cold,” with slight adjustments depending on the database. No restrictions were placed on language, publication date, or publication status. We also screened the references of articles and published systematic reviews for additional relevant studies. 

The article was included if the following inclusion criteria were met: (1) conducted in humans, (2) investigated the clinical effect of green tea or catechin in preventing influenza infection, and (3) included a control group. Two authors (Anchalee Rawangkan (A.R.) and Surasak Saokaew (S.S.)) independently reviewed all titles and abstracts to clarify whether the studies evaluated the clinical effects of green tea or catechin according to the inclusion criteria, and then analyzed data from potential full-text articles. When inconsistencies and instabilities regarding qualification arose, they were resolved by consensus.

### 4.2. Data Extraction and Outcome Measures

Two investigators (A.R. and S.S.) extracted the data, which were confirmed by Kirati Kengkla (K.K.) and Sukrit Kanchanasurakit (S.K.). The data extraction form included the following information: study design, duration, number of participants, participant age, participant sex, route of administration (gargling, drinking or taking capsules), concentration of GTCs consumed per day, outcome measurement, type of influenza virus infection, and influenza vaccine history of the participant. Outcomes of interest depended on influenza prevention; for example, laboratory-confirmed antigen tests of influenza viruses showed negative results. 

### 4.3. Quality Assessment 

Two investigators (A.R. and K.K.) performed quality assessment of each study using the Cochrane Collaboration Risk of Bias tool for randomized controlled trials (RoB 2.0) [59]. The Risk Of Bias In Non-randomized Studies of Interventions (ROBINS-I) tool [60] and the Newcastle–Ottawa Scale (NOS) [61] were used for observational studies. The six domains of the Cochrane tool assessed selection bias, performance bias, detection bias, attrition bias, reporting bias, and other sources of bias. Each domain of an RCT study was classified as unclear, low, or high risk. ROBINS-I, a new tool for assessing the risk of bias in estimates of comparative effectiveness of interventions from studies that did not use randomization to assign units to comparison groups, was assessed using the following seven domains: confounding domain, selection of participants into the study, classification of interventions, deviation from intended interventions, missing data, measurement of outcomes, and selection of reported outcomes. The NOS performed a star rating system for the qualification of the three domains, i.e., selection of study groups, comparability of groups, and coverage of the outcome of interest. Each domain of the NOS was rated as unclear (no star), low risk (one or more stars), or high risk (no star), for a total of nine stars, with studies reporting a total score of 7 being defined as high quality. Differences between analysts were resolved through discussion and agreement as to the final decision.

### 4.4. Statistical Analysis

Risk ratio (RR), a ratio of the risk of influenza among green tea and non-green tea groups, were estimated with 95% confidence intervals (CIs) for dichotomous outcomes. A risk ratio of 1.0 indicates identical risk among the two groups. RR > 1.0 indicates an increased risk of influenza in the green tea group, and RR < 1.0 indicates a decreased risk of influenza in the green tea group. In addition, standardized mean differences (SMDs) with 95% CIs were used for continuous outcomes. Heterogeneity was assessed using the Cochrane Q statistic and quantified using the *I*^2^ statistic [62]. A random-effects model was used to assess pooled RRs and 95% CIs [63,64]. We then assessed small-study effects using Begg’s test, Egger’s regression asymmetry test, and visual inspection of the funnel plot, whose asymmetric shape indicates the presence of bias [65]. Subgroup and sensitivity analyses were performed to examine the influence of each variable based on the baseline characteristics of each study, including: 1) models (fixed effect model and random effect model); 2) dose of epigallocatechin gallate (EGCG); 3) route of administration; 4) type of influenza virus; 5) history of participants with respect to the influenza vaccine (more than 80 percent of participants having received vaccinations); and 6) participant age. Articles with high risk of bias or low quality of evidence (NOS less than 7 stars) were omitted. Statistical tests were two-sided, with *p*-values of 0.05 indicating statistical significance.

## 5. Conclusions

This meta-analysis shows that regular green tea consumption, whether by taking GTC capsules, drinking, or gargling, can prevent influenza, although the study populations were from different countries, the treatment regimens were different, and the number of studies on tea catechins against influenza was limited. Further studies are needed to better investigate the processes described for individual and collective treatment with green tea. Future large-scale studies are needed to establish or confirm their clinical efficacy. Most importantly, it must be clearly emphasized that green tea catechins cannot replace standard influenza vaccination or treatment. Nevertheless, their beneficial effects may support common influenza prevention.

## Figures and Tables

**Figure 1 molecules-26-04014-f001:**
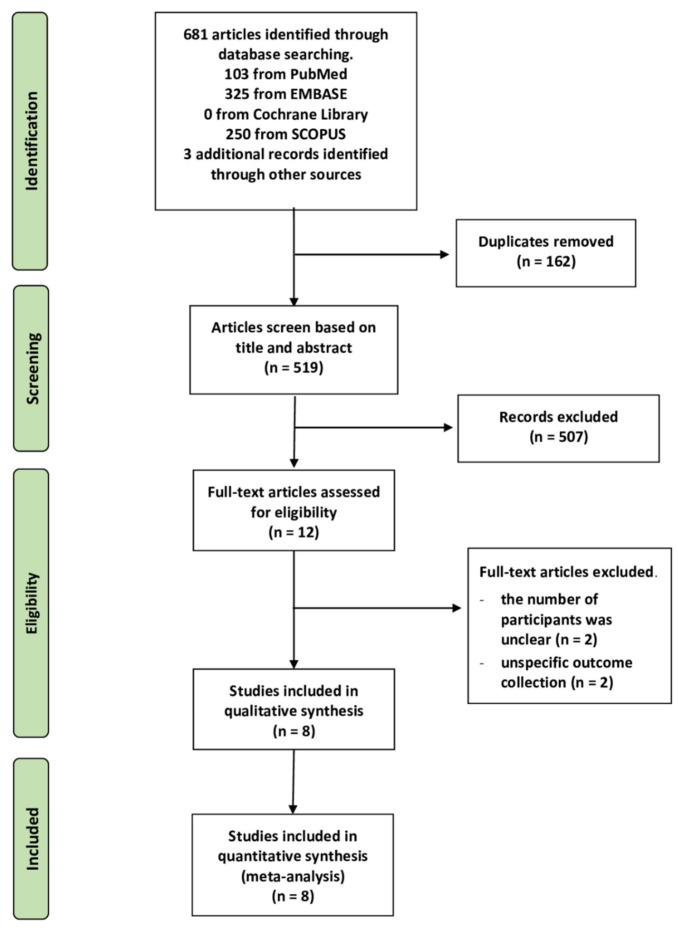
PRISMA flow diagram summary of the study selection process.

**Figure 2 molecules-26-04014-f002:**
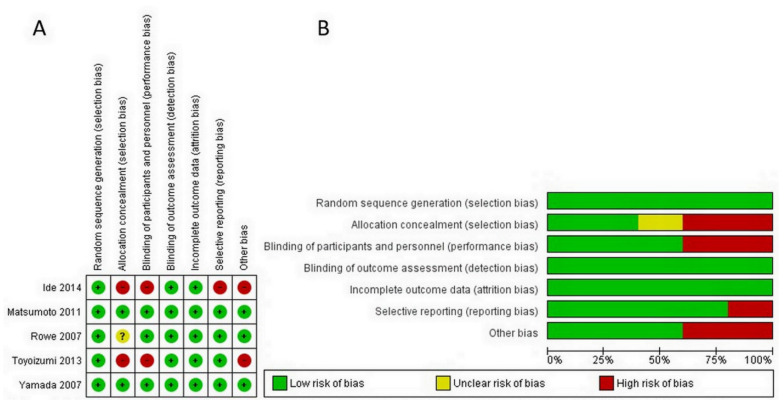
Assessment of risk of bias for RCTs. (**A**) Risk of bias graph showing each item presented as percentages across all RCT studies (**B**) Risk of bias summary showing each item for each study.

**Figure 3 molecules-26-04014-f003:**
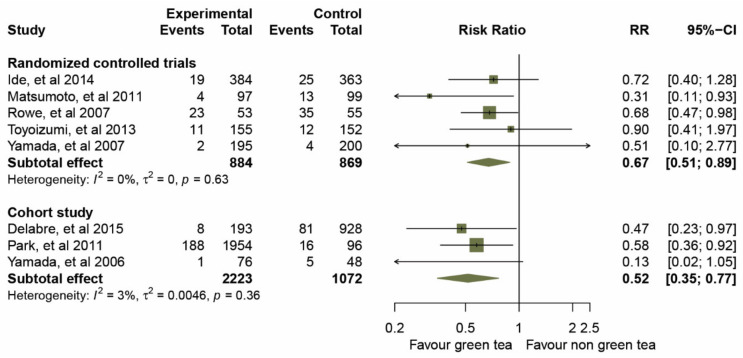
Effect of green tea on preventative influenza virus infection determined by subgroup analysis of RCTs and cohort studies.

**Figure 4 molecules-26-04014-f004:**
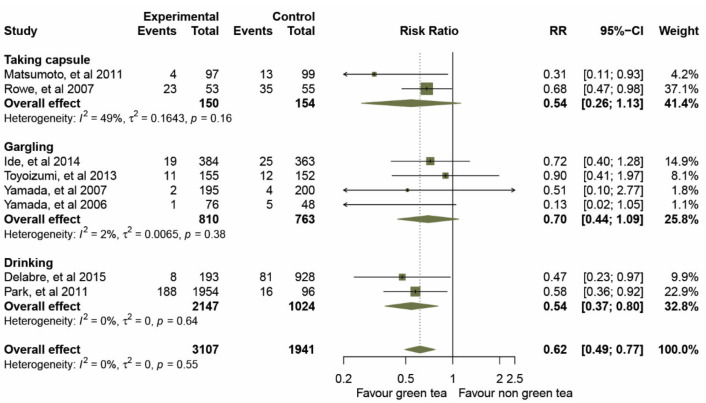
Effect of green tea on preventing influenza virus infection determined by route of administration.

**Table 1 molecules-26-04014-t001:** Characteristics of the included studies.

Authors (Year)	Study Design	Duration of Study (Months)	Route of Administration	Experimental Intervention GTCs mg/day	Age (Years)	Sex (M/F)	Received Influenza Vaccination			Clinical Outcome				
							Yes	No	N/A	Flu+	Flu−	Flu+	Flu−	Total
Yamada et al. (2006) [37]	Cohort	3	Gargling	100 mg By 200 µg/mL ×3 times, 500 mL	65–83	40/84	124	0	0	1	75	5	43	124
Rowe et al. (2007) [32]	RCT	3	Taking capsules	1500 mg By 2 capsules *	18–70	43/65	0	108	0	23	30	35	20	108
Yamada et al. (2007) [33]	RCT	3	Gargling	120 mg By 400 µg/mL ×3 times, 300 mL	20–65	NA	395	0	0	2	193	4	196	395
Matsumoto et al. (2011) [34]	RCT	5	Taking capsules	378 mg By 63 mg ×6 capsules	21–69	44/152	182	14	0	4	93	13	86	196
Park et al. (2011) [38]	Cohort	4	Drinking	137–685 mg By 1–5 cups/day	6–13	991/1059	1141	854	55	188	1766	16	80	2050
Toyoizumi et al. (2013) [35]	RCT	3	Gargling	280 mg By 560 µg/mL ×3 times, 500 mL	15–20	184/124	130	177	0	11	144	12	140	307
Ide et al. (2014) [36]	RCT	3	Gargling	185 mg By 370 µg/mL ×3 times, 500 mL	15–17	423/324	197	550	0	19	365	25	338	747
Delabre et al. (2015) [39]	Cohort	8	Drinking	300 mg By ≤2 cups/week	15–50	520/601	0	1121	0	8	185	81	847	1121

Abbreviations: N/A, not available, * approximately 10 cups/day; 120 mL/cup contains catechins 150 mg [10].

**Table 2 molecules-26-04014-t002:** Overall scientific quality of the cohort studies based on the Newcastle–Ottawa Scale (NOS).

Study Criterion	Yamada et al. (2006) [37]	Park et al. (2011) [38]	Delabre et al. (2015) [39]
Selection (maximum ****)	****	***	****
Comparability (maximum **)	**	*	*
Outcome (maximum ***)	***	**	***

Note: The symbol * indicates the number corresponding item applies to the study.

**Table 3 molecules-26-04014-t003:** Sensitivity and subgroup analysis.

Characteristics	All Studies			Randomized Control Trials			Cohort Studies		
	Risk Ratio (95% CIs)	Heterogeneity		Risk Ratio (95% CIs)	Heterogeneity		Risk Ratio (95% CIs)	Heterogeneity	
		*I*^2^-Index	*p*-Value		*I*^2^-Index	*p*-Value		*I*^2^-Index	*p*-Value
**Models**									
∙ Fixed effects model	0.59 (0.47. 0.74)	0.0%	0.55	0.66 (0.50, 0.88)	0.0%	0.63	0.49 (0.33, 0.73)	3.0%	0.36
∙ Random effects model	0.62 (0.49. 0.77)	0.0%	0.55	0.67 (0.51, 0.89)	0.0%	0.63	0.52 (0.35, 0.77)	3.0%	0.36
**Omission of Ide, et al. and Toyoizumi, et al. in the analysis of randomized control trials ^a^**									
∙ Before omission	0.62 (0.49. 0.77)	0.0%	0.55	0.67 (0.51, 0.89)	0.0%	0.63	0.52 (0.35, 0.77)	3.0%	0.36
∙ After omission	0.58 (0.45, 0.74)	0.0%	0.45	0.62 (0.43, 0.88)	1.0%	0.36	0.52 (0.35, 0.77)	3.0%	0.36
**Omission of Park, et al. in the analysis of cohort study ^b^**									
∙ Before omission	0.62 (0.49. 0.77)	0.0%	0.55	0.67 (0.51, 0.89)	0.0%	0.63	0.52 (0.35, 0.77)	3.0%	0.36
∙ After omission	0.63 (0.49, 0.81)	0.0%	0.44	0.67 (0.51, 0.89)	0.0%	0.63	0.36 (0.13, 1.03)	26.0%	0.24
**Dose of EGCG (mg/day)**									
∙ > 338 mg/day	0.68 (0.47, 0.98)	N/A	N/A	0.68 (0.47, 0.98)	N/A	N/A	N/A	N/A	N/A
∙ ≤ 338 mg/day	0.58 (0.44, 0.77)	0.0%	0.50	0.66 (0.44, 1.01)	0.0%	0.46	0.52 (0.35, 0.77)	3.0%	0.36
**Route of administration**									
∙ Gargling	0.70 (0.44, 1.09)	2.0%	0.38	0.75 (0.48, 1.18)	0.0%	0.81	0.13 (0.02 1.05)	N/A	N/A
∙ Taking capsules	0.54 (0.26, 1.13)	49.0%	0.16	0.54 (0.26, 1.13)	49.0%	0.16	N/A	N/A	N/A
∙ Drinking	0.54 (0.37, 0.80)	0.0%	0.64	N/A	N/A	N/A	0.54 (0.37, 0.80)	0.0%	0.64
**Type of Influenza virus**									
∙ Type A	0.51 (0.10, 2.77)	N/A	N/A	0.51 (0.10, 2.77)	N/A	N/A	N/A	N/A	N/A
∙ Type B	0.13 (0.02 1.05)	N/A	N/A	N/A	N/A	N/A	0.13 (0.02 1.05)	N/A	N/A
∙ Type A or B	0.63 (0.50, 0.79)	0.0%	0.62	0.68 (0.51, 0.90)	0.0%	0.48	0.54 (0.37, 0.80)	0.0%	0.64
**Received influenza vaccinations (more than 80 percent of participants)**									
∙ Yes	0.31 (0.13, 0.71)	0.0%	0.59	0.36 (0.15, 0.90)	0.0%	0.63	0.13 (0.02 1.05)	N/A	N/A
∙ No	0.65 (0.52, 0.82)	0.0%	0.76	0.72 (0.54, 0.96)	0.0%	0.82	0.54 (0.37, 0.80)	0.0%	0.64
**Age over 65 years**									
∙ Yes	0.13 (0.02 1.05)	N/A	N/A	N/A	N/A	N/A	0.13 (0.02 1.05)	N/A	N/A
∙ No	0.63 (0.50, 0.79)	0.0%	0.73	0.67 (0.51, 0.89)	0.0%	0.63	0.54 (0.37, 0.80)	0.0%	0.64

Abbreviations: CIs, confidence intervals; N/A, not available; EGCG, epigallocatechin gallate. ^a^ Selected for high risk of bias; ^b^ Selected from Newcastle–Ottawa Scale (less than 7 stars).

## Data Availability

The data presented in this study are available in this article.

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
