# Peer review of "Anti-Influenza with Green Tea Catechins: A Systematic Review and Meta-Analysis"

_molecules, 2021, doi:10.3390/molecules26134014_

Round 1
Reviewer 1 Report
Manuscript ID: molecules-1278243
Reviewer comment:
- The manuscript entitled ,,Anti-influenza with green tea catechins: a systematic review and meta-analysis ‘’ provides interesting data on this topic. The work is well described and the results interesting.
This meta-analysis adhered to the standard methodology of systematic reviews and meta-analyses as required by the PRISMA checklist (Supplementary data). Finally, study covered the current evidence and was conducted using appropriate statistical methods for the analyses. It shows that regular green tea consumption can prevent influenza by taking GTCs capsules, drinking or gargling.
This report may be an additional preventive measure for the prevention of influenza for the common person and may be regarded as an alternative to a pharmaceutical strategy. Nevertheless, to confirm the findings before implementation, longitudinal clinical trials with specific doses of green tea consumption are warranted.The topic of this study is interesting and in my opinion it could be interesting for a reasonable number of scientists and ordinary people during the flu epidemic. But further studies are needed to better investigate the processes described for individual and collective green tea treatment-related.
I suggest acceptance of this paper.
Author Response
We would like to thank the editor and reviewers for careful and thorough review of this manuscript. We have revised our manuscript in response to your suggestions point-by-point and the changes have made using “track change”. We hope that this improved manuscript is acceptable for publication in Molecules. The answer to their specific comments/suggestions showed in attached file.

Reviewer 2 Report
x
- Introduction: Mention general benefits from catechins: antioxidant, anti-inflammatory, anti-cancer effects…
- Also mention all catechin derivatives in green tea (EGCG, ECG, EGC, and EC) and other major constituents vitamins (A, B, C, E, K)
Musial, C., Kuban-Jankowska, A., & Gorska-Ponikowska, M. (2020). Beneficial properties of green tea catechins. International journal of molecular sciences, 21(5), 1744.
- Line 59: Please compare the stability and bioavailability of EGCG in the green tea extracts and pure epigallocatechin gallate
Fernández, V. A., Almeida Toledano, L., Pizarro Lozano, N., Tapia, E. N., Gómez Roig, M. D., De la Torre Fornell, R., & García Algar, Ó. (2020). Bioavailability of epigallocatechin gallate administered with different nutritional strategies in healthy volunteers. Antioxidants, 9(5), 440.
Yong Feng, W. (2006). Metabolism of green tea catechins: an overview. Current drug metabolism, 7(7), 755-809.
Zhu, Q. Y., Zhang, A., Tsang, D., Huang, Y., & Chen, Z. Y. (1997). Stability of green tea catechins. Journal of Agricultural and Food Chemistry, 45(12), 4624-4628.
- Line 62: Please explain that EGCG and ECG exhibited hemagglutination inhibition activity (Song et al 2005)
Song, J.-M.; Lee, K.-H.; Seong, B.-L. Antiviral effect of catechins in green tea on influenza virus. Antiviral Research 2005, 68, 66-145 74, doi:https://doi.org/10.1016/j.antiviral.2005.06.010.
- Line 80: Define Risk ratio
- Lines 66-78: Please, provide more detailed explanations of the principles of ROBINS-I and NOS tools for observation studies
- Line 95: Expand the conclusion chapter with the main conclusions from 3.1 and 3.2 sections
Author Response

(The authors gave the same response as above.)

Reviewer 3 Report
According to my information, the manuscript under review, is the first one related with the antivirals area. In general, the approach applied by the authors – systematic review of literary data and meta- analysis, represents interest because demonstrates a panoramic view on one special topic – anti-flu effect of green tea catechins. Having in mind the place of the green tea in the nutrition in global scale, the importance of such study has to be underlined. The summary data collection of eight studies embracing 5.048 participants is very impressive. The final conclusions formulated that (1) green tea possesses anti-flu activity and (2) green tea consumption has additional preventive measure for influenza prevention, merit a particular attention. Comparison of the data of three ways of green tea application – drinking, gargling and taking capsules, enlarged to importance of the study. Authors characterised and discussed intensively the most probable limitations of that statements.
As virologists working in the field of experimental chemotherapy of viral infections, flu included, I have not experience in meta-analysis, statistical methodology used, but it is not very acceptable the combination of data of studies carried out with sharply great differences between studied groups of patients.
Author Response

(The authors gave the same response as above.)

Round 2
Reviewer 2 Report
The paper entitled: „Anti-influenza with green tea catechins: a systematic review 2 and meta-analysis“ written by Rawangkan et al., is well organized and written paper. After revision, the paper is now suitable for publication.